# ARCHITECT OF THE BITS WORLD: MASKED AUTOREGRESSIVE MODELING FOR CIRCUIT GENERATION GUIDED BY TRUTH TABLE

## ABSTRACT

Logic synthesis, a critical stage in electronic design automation (EDA), optimizes gate-level circuits to minimize power consumption and area occupancy in integrated circuits (ICs). Traditional logic synthesis tools rely on human-designed heuristics, often yielding suboptimal results. Although differentiable architecture search (DAS) has shown promise in generating circuits from truth tables, it faces challenges such as high computational complexity, convergence to local optima, and extensive hyperparameter tuning. Consequently, we propose a novel approach integrating conditional generative models with DAS for circuit generation. Our approach first introduces CircuitVQ, a circuit tokenizer trained based on our Circuit AutoEncoder We then develop CircuitAR, a masked autoregressive model leveraging CircuitVQ as the tokenizer. CircuitAR can generate preliminary circuit structures from truth tables, which guide DAS in producing functionally equivalent circuits. Notably, we observe the scalability and emergent capability in generating complex circuit structures of our CircuitAR models. Extensive experiments also show the superior performance of our method. This research bridges the gap between probabilistic generative models and precise circuit generation, offering a robust solution for logic synthesis.

## 1 INTRODUCTION

With the rapid advancement of technology, the scale of integrated circuits (ICs) has expanded exponentially. This expansion has introduced significant challenges in chip manufacturing, particularly concerning power and area metrics. A primary objective in IC design is achieving the same circuit function with fewer transistors, thereby reducing power usage and area occupancy.

Logic synthesis (Hachtel & Somenzi, 2005a), a critical step in electronic design automation (EDA), transforms behavioral-level circuit designs into optimized gate-level circuits, ultimately yielding the final IC layout. The primary goal of logic synthesis is to identify the physical implementation with the fewest gates for a given circuit function. This task constitutes a challenging NP-hard combinatorial optimization problem (Hachtel & Somenzi, 2005b). Current logic synthesis tools (Brayton & Mishchenko, 2010; Wolf et al., 2013) rely on human-designed heuristics, often leading to suboptimal outcomes.

Differentiable architecture search (DAS) techniques (Liu et al., 2018; Chu et al., 2020) offer novel perspectives on addressing challenges in this problem. Circuit functions can be represented through truth tables, which map binary inputs to their corresponding outputs. Truth tables provide a precise representation of input-output relationships, ensuring the design of functionally equivalent circuits. Inspired by this, researchers (Hillier et al., 2023b; Wang et al., 2024) have begun exploring the application of DAS to synthesize circuits directly from truth tables. Specifically, Hillier et al. (2023b) proposed CircuitNN, a framework that learns differentiable connection structures with logic gates, enabling the automatic generation of logic circuits from truth tables. This approach significantly reduces the complexity of traditional circuit generation. Building on this, Wang et al. (2024) introduced T-Net, a triangle-shaped variant of CircuitNN, incorporating regularization techniques to enhance the efficiency of DAS.

Despite these advancements, several challenges remain. The computational complexity of DAS grows quadratically with the number of gates, posing scalability issues. Although triangle-shaped architecture (Wang et al., 2024) partially mitigates this problem, redundancy persists. Additionally, DAS is susceptible to converging to local optima (Liu et al., 2018), where network depth and layer width require extensive searches. The challenges arise from the vast search space in DAS. Intuitively, limiting the search space through predefined parameters, including network depth, gates per layer, and connection probabilities, can significantly reduce the complexity.

Recent advances (OpenAI, 2023; Abramson et al., 2024; Esser et al., 2024; Li et al., 2024a) in conditional generative models have demonstrated remarkable performance across language, vision, and graph generation tasks. Motivated by these developments, we propose a novel approach to circuit generation that generates preliminary circuit structures to guide DAS in generating refined circuits matching specified truth tables. Firstly, we introduce CircuitVQ, a tokenizer with a discrete codebook for circuit tokenization. Built upon our Circuit AutoEncoder framework (Hou et al., 2022; Li et al., 2023a; Wu et al., 2025), CircuitVQ is trained through a circuit reconstruction task. Specifically, the CircuitVQ encoder encodes input circuits into discrete tokens using a learnable codebook, while the decoder reconstructs the circuit adjacency matrix based on these tokens. Subsequently, the CircuitVQ encoder serves as a circuit tokenizer for CircuitAR pretraining, which employs a masked autoregressive modeling paradigm (Chang et al., 2022; Li et al., 2023b). In this process, the discrete codes function as supervision signals. After training, CircuitAR can generate discrete tokens progressively, which can be decoded into initial circuit structures by the decoder of CircuitVQ. These prior insights can guide DAS in producing refined circuits that match the target truth tables precisely. Our key contributions can be summarized as follows:

- We introduce CircuitVQ, a circuit tokenizer that facilitates graph autoregressive modeling for circuit generation, based on our Circuit AutoEncoder framework;
- Develop CircuitAR, a model trained using masked autoregressive modeling, which generates initial circuit structures conditioned on given truth tables;
- Propose a refinement framework that integrates differentiable architecture search to produce functionally equivalent circuits guided by target truth tables;
- Comprehensive experiments demonstrating the scalability and capability emergence of our CircuitAR and the superior performance of the proposed circuit generation approach.

## 2 PRELIMINARIES

### 2.1 MODELING CIRCUIT AS DAG

In this work, we model the circuit as a directed acyclic graph (DAG) (Brummayer & Biere, 2006), which facilitates graph autoregressive modeling. Specifically, each node in the DAG corresponds to a logic gate, while the directed edges represent the connections between these components.

### 2.2 DIFFERENTIABLE CIRCUITNN

As depicted in Figure 1, CircuitNN (Hillier et al., 2023b) replaces traditional neural network layers with logic gates (e.g., NAND) as basic computational units, learning to synthesize circuits by optimizing logic correctness based on truth tables. During training, input connections of each gate are determined through learnable probability distributions, enabling adaptive circuit architecture modification. To enable gradient-based learning, CircuitNN transforms discrete logic operations into continuous, differentiable functions using NAND gates for simplicity. The NAND gate is logically complete, allowing the construction of any complex logic circuit. Its continuous relaxation can be defined as:

$$\text{NAND}(x, y) = 1 - x \cdot y, \text{ where } x, y \in [0, 1]. \tag{1}$$

Additionally, CircuitNN employs Gumbel-Softmax (Jang et al., 2016) for stochastic sampling of gate inputs. Through stochastic relaxation, gate and network outputs are no longer binary but take continuous values ranging from 0 to 1 instead. This end-to-end differentiability allows the model to learn gate input distributions using gradient descent. After training, the continuous, probabilistic circuit is converted back into a discrete logic circuit by selecting the most probable connections based on the learned probability distributions, as shown in Figure 1.

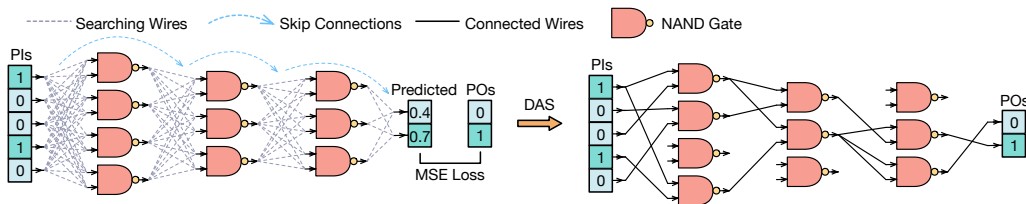

Figure 1: Illustration of differentiable CircuitNN.

## 3 METHODOLOGY

In this section, we first introduce CircuitVQ (Section 3.2), a model built upon the Circuit AutoEncoder framework (Section 3.1) and trained with the task of circuit reconstruction. Utilizing CircuitVQ as a tokenizer, we subsequently train CircuitAR (Section 3.3) with graph autoregressive modeling paradigm, which can generate preliminary circuit structures conditioned on a provided truth table. Finally, the initial circuit structure generated by CircuitAR serves as a guide for DAS (Section 3.4) to refine and generate circuits functionally equivalent to the given truth table.

### 3.1 CIRCUIT AUTOENCODER

Let $\mathcal{G} = (\mathcal{V}, \mathcal{A})$ represent a circuit, where $\mathcal{V}$ denotes the set of $N$ nodes, with each node $v_i \in \mathcal{V}$. Following the architecture of CircuitNN (Hillier et al., 2023b), each node $v_i$ can be classified into one of three types: primary inputs (PIs), primary outputs (POs), and NAND gates, each labeled by $u_i \in \mathcal{U}, i \in \{1, 2, 3\}$ respectively. The adjacency matrix $\mathcal{A} \in \{0, 1\}^{N \times N}$ captures the connectivity between nodes, where $\mathcal{A}_{i,j} = 1$ indicates the presence of a directed edge from $v_i$ to $v_j$. In the circuit autoencoder framework, an encoder, denoted as $g_E$, encodes the circuit $\mathcal{G}$ into a latent representation $\boldsymbol{Z} \in \mathbb{R}^{N \times d}$ with dimensionality $d$. The encoding process for a circuit can be formulated as:

$$\boldsymbol{Z} = g_E(\mathcal{V}, \mathcal{A}). \tag{2}$$

Simultaneously, a decoder $g_D$ aims to reconstruct the original circuit $\mathcal{G}$ from the latent representation $\boldsymbol{Z}$. Since node types can be directly derived from the truth table, the decoder is designed to focus on reconstructing the adjacency matrix $\mathcal{A}$, which can be formalized as follows:

$$\tilde{\mathcal{G}} = (\mathcal{V}, \tilde{\mathcal{A}}) = (\mathcal{V}, f(g_D(\boldsymbol{Z}, \mathcal{V}))), \tag{3}$$

where $\tilde{\mathcal{A}} \in \mathbb{R}^{N \times N}$ denotes the reconstructed adjacency matrix, obtained by decoding $\boldsymbol{Z}$ through $g_D$ and applying a mapping function $f : \mathbb{R}^{N \times d} \to \mathbb{R}^{N \times N}$. Meanwhile, $\tilde{\mathcal{G}}$ represents the reconstructed graph. A robust encoder $g_E$ capable of capturing fine-grained structural information is essential to facilitate the circuit reconstruction task. We incorporate the Graphormer (Ying et al., 2021) architecture into $g_E$. For the decoder $g_D$, we adopt a simple Transformer architecture.

### 3.2 CIRCUITVQ

As mentioned in Section 3.1, we propose a circuit autoencoder architecture for the circuit reconstruction task. The outputs of $g_E$ and the inputs of $g_D$ are continuous. The circuit tokenizer is required to map the circuit to a sequence of discrete circuit tokens for masked autoregressive modeling, illustrated in Section 3.3. Specifically, a circuit $\mathcal{G}$ can be tokenized to $\boldsymbol{Y} = [y_1, y_2, \cdots, y_N] \in \mathbb{R}^N$ using the circuit quantizer $\mathcal{C}$ which contains $K$ discrete codebook embeddings. Here, each token $y_i$ belongs to the vocabulary set $\{1, 2, \ldots, K\}$ of $\mathcal{C}$. Consequently, we develop a circuit tokenizer, CircuitVQ, based on the circuit autoencoder by integrating a circuit quantizer $\mathcal{C}$. As shown in Figure 2, the tokenizer comprises three components: a circuit encoder $g_E$, a circuit quantizer $\mathcal{C}$, and a circuit decoder $g_D$.

Firstly, $g_E$ encodes the circuit into vector representations $\boldsymbol{Z}$. Subsequently, $\mathcal{C}$ identifies the nearest neighbor in the codebook for $z_i \in \boldsymbol{Z}$. Let $\{e_1, e_2, \ldots, e_K\}$ represent the codebook embeddings and $e_K \in \mathbb{R}^d$. For the $i$-th node, the quantized code $y_i$ is determined by:

$$y_i = \arg\min_j ||\ell_2(\boldsymbol{z}_i) - \ell_2(\boldsymbol{e}_j)||_2, \tag{4}$$

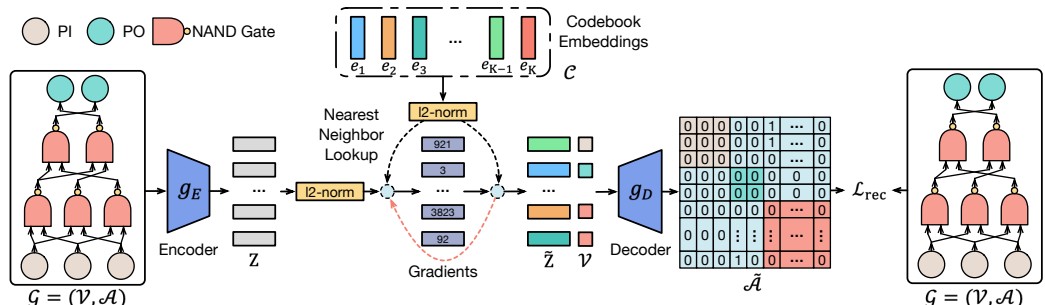

Figure 2: The training process of CircuitVQ.

where $j \in \{1, 2, \ldots, K\}$ and $\ell_2$ normalization $\ell_2(\boldsymbol{v}) = \frac{\boldsymbol{v}}{\|\boldsymbol{v}\|_2}$ is applied during the codebook lookup (Yu et al., 2021). This distance metric is equivalent to selecting codes based on cosine similarity. Consequently, the output of $\mathcal{C}$ for each node representation $\boldsymbol{z}_i$ the can be calculated based on the given Equation (4):

$$\tilde{\boldsymbol{z}}_i = \mathcal{C}(\boldsymbol{z}_i) = \ell_2(\boldsymbol{e}_{y_i}), \text{ where } \tilde{\boldsymbol{z}}_i \in \tilde{\boldsymbol{Z}}. \quad (5)$$

After quantizing the circuit into discrete tokens, the $\ell_2$-normalized codebook embeddings $\tilde{\boldsymbol{Z}} = \{\tilde{\boldsymbol{z}}_i\}_{i=1}^N$ are fed to $g_D$. The output vectors $\tilde{\boldsymbol{X}} = \{\tilde{\boldsymbol{x}}_i\}_{i=1}^N = g_D(\tilde{\boldsymbol{Z}}, \mathcal{V})$ are used to reconstruct the original adjacency matrix $\mathcal{A}$ of the circuit $\mathcal{G}$.

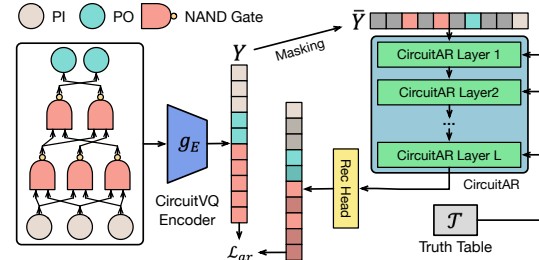

Figure 3: The training process of CircuitAR under the condition of the truth table, leveraging CircuitVQ as the tokenizer.

Specifically, the reconstructed adjacency matrix $\tilde{\mathcal{A}}$ is derived from the output vectors $\tilde{\boldsymbol{X}}$ as follows:

$$\tilde{\mathcal{A}} = f(\tilde{\boldsymbol{X}}) = \sigma\left(f_1(\tilde{\boldsymbol{X}}) \cdot f_2(\tilde{\boldsymbol{X}})^\top\right), \quad (6)$$

where both $f_1 : \mathbb{R}^d \to \mathbb{R}^d$ and $f_2 : \mathbb{R}^d \to \mathbb{R}^d$ are learnable projection functions, and $\sigma(x)$ denotes the sigmoid function. The training objective of the circuit reconstruction task is to minimize the binary cross-entropy loss between the reconstructed adjacency matrix $\tilde{\mathcal{A}}$ and the original adjacency matrix $\mathcal{A}$, which can be calculated as follows:

$$\mathcal{L}_{\text{rec}} = -\frac{1}{N^2} \sum_{i=1}^N \sum_{j=1}^N \left[\mathcal{A}_{ij} \log(\tilde{\mathcal{A}}_{ij}) + (1 - \mathcal{A}_{ij}) \log(1 - \tilde{\mathcal{A}}_{ij})\right]. \quad (7)$$

Given that the quantization process in Equation (4) is non-differentiable, gradients are directly copied from the decoder input to the encoder output during backpropagation, which enables the encoder to receive gradient updates. Intuitively, while the quantizer selects the nearest codebook embedding for each encoder output, the gradients of the codebook embeddings provide meaningful optimization directions for the encoder. Consequently, the overall training loss for CircuitVQ is:

$$\mathcal{L}_{\text{vq}} = \mathcal{L}_{\text{rec}} + \|\boldsymbol{Z} - \text{sg}[\boldsymbol{E}]\|_2^2 + \beta \cdot \|\text{sg}[\boldsymbol{Z}] - \boldsymbol{E}\|_2^2, \quad (8)$$

where $\text{sg}[\cdot]$ stands for the stop-gradient operator, which is an identity at the forward pass while having zero gradients during the backward pass. $\boldsymbol{E} = \{\boldsymbol{e}_{y_i}\}_{i=1}^N$ and $\beta$ denotes the hyperparameter for commitment loss (Van Den Oord et al., 2017).

### 3.3 CIRCUITAR

After completing the CircuitVQ training, we train CircuitAR using a graph autoregressive modeling paradigm as shown in Figure 3, where CircuitVQ functions as the tokenizer. Let $\boldsymbol{Y} = [y_i]_{i=1}^N$ represent the discrete latent tokens of the input circuit $\mathcal{G}$, tokenized by CircuitVQ. During the masked

autoregressive training process, we sample a subset of nodes $\mathcal{V}_s \subset \mathcal{V}$ and replace them with a special mask token $m$. For the masked $\boldsymbol{Y}$, the latent token $\bar{y}_i$ is defined as:

$$\bar{y}_i = \begin{cases} y_i, & \text{if } v_i \notin \mathcal{V}_s; \\ m, & \text{if } v_i \in \mathcal{V}_s. \end{cases} \tag{9}$$

Following Chang et al. (2022) and Li et al. (2024a), we employ a cosine mask scheduling function $\gamma(r) = \cos(0.5\pi r)$ in the sampling process. This involves uniformly sampling a ratio $r$ from the interval $[0, 1]$ and then selecting $\lceil \gamma(r) \cdot N \rceil$ tokens from $\boldsymbol{Y}$ to mask uniformly. Let $\bar{\boldsymbol{Y}} = [\bar{y}_i]_{i=1}^N$ denote the output after applying the masking operation to $\boldsymbol{Y}$. The masked sequence $\bar{\boldsymbol{Y}}$ is then fed into a multi-layer transformer with bidirectional attention to predict the probabilities $p(y_i|\bar{\boldsymbol{Y}}, \mathcal{T})$ for each $v_i \in \mathcal{V}_s$ under the condition of the truth table. The transformer is designed based on Llama models, each CircuitAR layer consists of a self-attention block, a cross-attention block, and an FFN block. Specifically, the info of the truth table is conditioned by serving $\mathcal{T}$ as the input key and value of the cross-attention block. The training loss for CircuitAR is defined as:

$$\mathcal{L}_{\text{ar}} = -\sum_{\mathcal{D}} \sum_{v_i \in \mathcal{V}_s} \log p(y_i|\bar{\boldsymbol{Y}}, \mathcal{T}), \tag{10}$$

where $\mathcal{D}$ represents the set of training circuits.

**Autoregressive decoding.** We introduce a parallel decoding method, where tokens are generated in parallel. This approach is feasible due to the bidirectional self-attention mechanism of CircuitAR. At inference time, we begin with a blank canvas $\bar{\boldsymbol{Y}} = [m]^N$ and the decoding process of CircuitAR follows Algorithm 1. Specifically, the decoding algorithm generates a circuit in $T$ steps. At each iteration, the model predicts all tokens simultaneously but retains only the most confident predictions following the cosine schedule (Chang et al., 2022; Li et al., 2024a). The remaining tokens are masked and re-predicted in the next iteration. The mask ratio decreases progressively until all tokens are generated within $T$ iterations.

### 3.4 DIFFERENTIABLE ARCHITECTURE SEARCH

After completing the training process of CircuitAR, autoregressive decoding is performed based on the input truth table $\mathcal{T}$ to generate preliminary circuit structures represented by the reconstructed adjacency matrix $\tilde{\mathcal{A}}$. This matrix $\tilde{\mathcal{A}}$ can serve as prior knowledge for DAS, enabling the generation of a precise circuit that is logically equivalent to $\mathcal{T}$.

**DAG Search.** The reconstructed adjacency matrix $\tilde{\mathcal{A}}$ is a probability matrix that denotes the probabilities of connections between gates. However, $\tilde{\mathcal{A}}$ may contain cycles and identifying the optimal DAG with the highest edge probabilities is an NP-hard problem. Consequently, we employ a greedy algorithm to obtain a suboptimal DAG. As illustrated in Algorithm 2, the algorithm initializes $\bar{\mathcal{A}} \in \mathbb{R}^{N \times N}$ with edge probabilities and enforces basic structural rules: PIs have no indegree, POs have no outdegree, and self-loops are prohibited in circuit designs. Following this initialization, a depth-first search (DFS) is conducted to detect cycles in $\bar{\mathcal{A}}$. If no cycles are found, $\bar{\mathcal{A}}$ is a valid DAG, and the algorithm terminates. If a cycle is detected, the edge with the lowest probability within the cycle is identified and removed by setting the corresponding edge in $\bar{\mathcal{A}}$ to 0. This process repeats iteratively until no cycles remain. This greedy approach ensures the derivation of a valid DAG $\bar{\mathcal{A}}$ that approximates the optimal structure while preserving the acyclic property necessary for circuit design. The resulting DAG serves as a foundation for further refinement in the DAS process, ultimately generating a precise circuit that is logically equivalent to $\mathcal{T}$.

**Initialization.** After executing Algorithm 2, the adjacency matrix of a valid DAG $\bar{\mathcal{A}} \in \mathbb{R}^{N \times N}$ and its corresponding probability matrix $\hat{\mathcal{A}} = \bar{\mathcal{A}} \cdot \tilde{\mathcal{A}}$, where $\hat{\mathcal{A}} \in \mathbb{R}^{N \times N}$, are obtained. Using $\bar{\mathcal{A}}$, we derive the hierarchical structure $H = \{h_1, h_2, \ldots, h_l\}$, where $h_l$ represents the node list of the $l$-th layer. The set $H$ encapsulates the layer count $l$ and the width information of each layer, which is used to initialize CircuitNN illustrated in Figure 1. For connection probabilities, since each node can only connect to nodes from preceding layers, we normalize the connection probabilities such that their summation equals 1. This yields the weights $\boldsymbol{w} \in \mathbb{R}^{N_p}$ for possible connections, where $N_p$ denotes the number of nodes in the previous layer. To ensure compatibility with the Softmax function applied in CircuitNN, we initialize the logits $\hat{\boldsymbol{w}} \in \mathbb{R}^{N_p}$ such that the Softmax output

**Algorithm 1** Autoregressive Decoding

**Input:** Masked tokens $\bar{Y} = [\bar{y}_i]_{i=1}^N, \forall \bar{y}_i = m$, token length $N$, total iterations $T$.
**Output:** Predicted tokens $\tilde{Y} = [\tilde{y}_i]_{i=1}^N \forall \tilde{y}_i \neq m$.
1: **for** $t \leftarrow 0$ **to** $T-1$ **do**
2:   Initialize the number of masked tokens $n$;
3:   Compute probabilities $p(\bar{y}_i) \in \mathbb{R}^K$ for each $\bar{y}_i \in \bar{Y}$;
4:   Initialize $S \leftarrow [s_i]_{i=1}^N$, where $s_i = 0$, and $Y \leftarrow \bar{Y}$;
5:   **for** $i \leftarrow 1$ **to** $N$ **do**
6:     **if** $\bar{y}_i = m$ **then**
7:       Sample a token $o_i \in \{1, \ldots, K\}$ from $p(\bar{y}_i)$;
8:       $s_i \leftarrow p(\bar{y}_i)[o_i]$ and $\tilde{y}_i \leftarrow o_i$;
9:     **else**
10:       $s_i \leftarrow 1$;
11:     **end if**
12:   **end for**
13:   **for** $i \leftarrow 1$ **to** $N$ **and** $\bar{y}_i \neq m$ **do**
14:     $r \leftarrow$ sorted$(S)[n]$; // Select the $n$-th highest score from the sorted $S$ in decending order
15:     $\bar{y}_i \leftarrow \begin{cases} \tilde{y}_i, & \text{if } s_i < r, \\ \bar{y}_i, & \text{otherwise}; \end{cases}$
16:   **end for**
17:   $\tilde{Y} \leftarrow \bar{Y}$;
18: **end for**

**Algorithm 2** DAG Search

**Input:** Adjacency matrix $\tilde{\mathcal{A}}$, PI node list $Q_i$, PO node list $Q_o$.
**Output:** Adjacency matrix $\bar{\mathcal{A}}$ of a valid DAG.
1: Initialize $\bar{\mathcal{A}} \leftarrow \tilde{\mathcal{A}}$.
2: **for** each edge $(i, j)$ in $\tilde{\mathcal{A}}$ $(i \neq j)$ **do**
3:   $\bar{\mathcal{A}}[i][j] \leftarrow 0$.
4:   **if** $i \notin Q_o$ and $j \notin Q_i$ and $\tilde{\mathcal{A}}[i][j] > 0.5$ **then**
5:     $\bar{\mathcal{A}}[i][j] \leftarrow 1$;
6:   **end if**
7: **end for**
8: **while** True **do**
9:   $c \leftarrow$ cycleDetect$(\bar{\mathcal{A}})$; // Detect a cycle using DFS and return the list of nodes forming the cycle.
10:   **if** len$(c) = 0$ **then**
11:     **break**; // No cycles detected; $\bar{\mathcal{A}}$ is a valid DAG.
12:   **end if**
13:   Initialize $s \leftarrow \infty$.
14:   **for** $i \leftarrow 0$ **to** len$(c) - 1$ **do**
15:     $j \leftarrow c[i]$ and $k \leftarrow c[(i+1) \bmod \text{len}(c)]$;
16:     **if** $\tilde{\mathcal{A}}[j][k] < s$ **then**
17:       $s \leftarrow \tilde{\mathcal{A}}[j][k]$ and $r \leftarrow (j, k)$;
18:     **end if**
19:   **end for**
20:   $\bar{\mathcal{A}}[r[0]][r[1]] \leftarrow 0$;
21: **end while**

matches the normalized connection probabilities. The logits are initialized as follows:

$$\hat{w} = \log(w + \epsilon) - \frac{1}{N_p} \sum_{i=1}^{N_p} \log(w_i + \epsilon), \tag{11}$$

where $\epsilon$ is a small constant for numerical stability. After initialization, the precise circuit structure is obtained through DAS, guided by the input truth table. Notably, if DAS converges to a local optimum, the weights of the least initialized nodes can be randomly selected and reinitialized using Equation (11) to facilitate further optimization.

## 3.5 BITS DISTANCE

DAS introduces inherent randomness, complicating the evaluation of CircuitAR's circuit generation capability using post-DAS metrics. To overcome this, we introduce Bits Distance (BitsD), a metric offering a more reliable assessment of CircuitAR's conditional generation ability. BitsD quantifies the discrepancy between the outputs of an untrained CircuitNN, initialized via CircuitAR, and the labels from the truth table. It measures how well CircuitAR generates circuits conditioned on the truth table. Specifically, after initializing CircuitNN, we feed it with the truth table inputs and compute the mean absolute error (MAE) between the untrained CircuitNN outputs and the truth table labels. This MAE is defined as Bits Distance. A smaller BitsD indicates that the untrained CircuitNN is closer to the target circuit described by the truth table.

## 4 EXPERIMENTS

### 4.1 EXPERIMENT SETTINGS

**Data Augmentation.** We provide more details about data augmentation in Appendix D and investigate the impact of the idle of NAND gates in Appendix F.

**Training Details.** We generate a training dataset with around 400k circuits (average 200 gates per circuit) from the open-source datasets (Bryan, 1985; Albrecht, 2005; Amarú et al., 2015). The training dataset construction details will be illustrated in Appendix C. We also provide more details about the training processes of CircuitVQ and CircuitAR in Appendix B.1 and Appendix B.2.

**Baseline Selection.** For baseline selection, we choose CircuitNN (Hillier et al., 2023b) and T-Net (Wang et al., 2024) due to their state-of-the-art (SOTA) performance in circuit generation guided

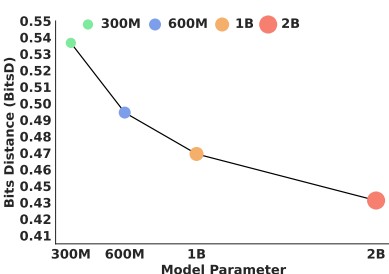

Figure 4: Scaling behavior of CircuitAR with different model parameters.

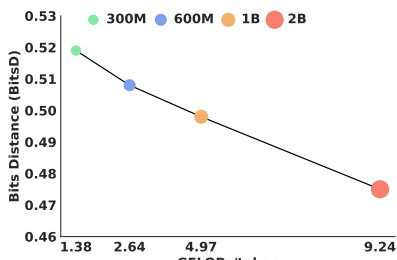

Figure 5: The performance of different model sizes under a fixed compute budget.

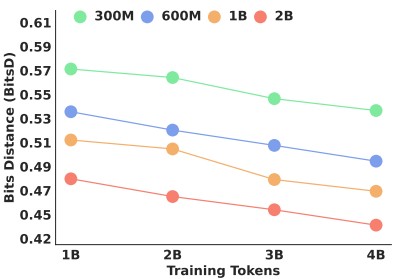

Figure 6: Training with more tokens improves BitsD for CircuitAR.

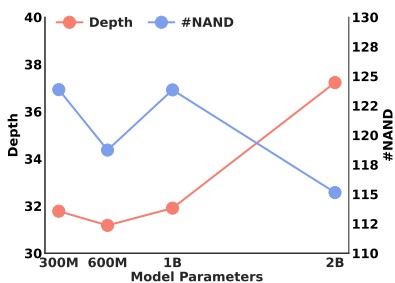

Figure 7: Emergent capability in generating complex circuit structures of our CircuitAR.

by truth tables. Additionally, several other studies (Tsaras et al., 2024; Li et al., 2024b; Zhou et al., 2024) have explored circuit generation using different paradigms. We discuss these approaches in Appendix A, as they diverge from the DAS paradigm employed in this work.

**Evaluation Details.** To validate the effectiveness of our CircuitAR models, we conduct evaluations using circuits from the IWLS competition (Mishchenko et al., 2022), which include five distinct function categories: random, basic functions, Espresso (Rudell, 1985), arithmetic functions, and LogicNets (Umuroglu et al., 2020). Random circuits consist of random and decomposable Boolean functions, basic functions include majority functions and binary sorters, and arithmetic functions involve arithmetic circuits with permuted inputs and dropped outputs. Furthermore, we evaluate the BitsD for CircuitAR models with different sizes to assess their conditional circuit generation capability. This evaluation is performed on our circuit generation benchmark with 1000 circuits separate from the training dataset.

## 4.2 SCALABILITY AND EMERGENT CAPABILITY

To analyze CircuitAR's scaling behavior, we perform experiments along two primary dimensions: parameter scaling (Figure 4) and data scaling (Figure 6). Our results reveal distinct performance patterns quantified through BitsD, demonstrating how these scaling axes influence performance. Additionally, we observe emergent capability in generating complex circuit structures of CircuitAR.

**Parameter Scaling.** As illustrated in Figure 4, increasing model capacity exhibits robust scaling laws. The 300M parameter model achieves 0.5317 BitsD, while scaling to 2B parameters yields 0.4362 BitsD. This progression follows a power-law relationship (Kaplan et al., 2020), where performance scales predictably with model size. Notably, marginal returns diminish at larger scales. The 0.3B→0.6B transition provides a 7.94% improvement versus 6.07% for 1B→2B, highlighting practical trade-offs between capacity and computational costs. These findings corroborate theoretical expectations (Thomas & Joy, 2006), confirming that larger models compress logical information more efficiently. Moreover, as shown in Figure 5, larger models achieve better performance under a fixed compute budget despite training on fewer tokens, owing to their superior capacity. Figure 5 also illustrates that BitsD scales inversely with the computing budget, which aligns with the scaling law (Kaplan et al., 2020) during LLM training.

Table 1: Experiment results of circuit generation accuracy and DAS steps. Impr. is the percentage decrease in DAS steps.

| Benchmark | | | | CircuitNN | | T-Net | | | CircuitAR-2B | | |
|---|---|---|---|---|---|---|---|---|---|---|---|
| Category | IWLS | # PI | # PO | Acc.(%)↑ | Steps↓ | Acc.(%)↑ | Steps↓ | Impr.(%)↑ | Acc.(%)↑ | Steps↓ | Impr.(%)↑ |
| Random | ex00 | 6 | 1 | 100 | 88715 | 100 | 85814 | 3.27 | 100 | 52023 | 41.36 |
| | ex01 | 6 | 1 | 100 | 64617 | 100 | 68686 | -6.30 | 100 | 29636 | 54.14 |
| Basic Functions | ex11 | 7 | 1 | 100 | 104529 | 100 | 49354 | 52.78 | 100 | 47231 | 54.81 |
| | ex16 | 5 | 5 | 100 | 115150 | 100 | 121108 | -5.17 | 100 | 45434 | 60.54 |
| | ex17 | 6 | 6 | 100 | 90584 | 100 | 57875 | 36.11 | 100 | 58548 | 35.66 |
| Expresso | ex38 | 8 | 7 | 100 | 86727 | 100 | 86105 | 0.71 | 100 | 74847 | 13.70 |
| | ex46 | 5 | 8 | 100 | 75726 | 100 | 75603 | 0.16 | 100 | 26854 | 64.54 |
| Arithmetic Function | ex50 | 8 | 2 | 100 | 87954 | 100 | 65689 | 25.31 | 100 | 42729 | 51.42 |
| | ex53 | 8 | 2 | 100 | 92365 | 100 | 75140 | 18.65 | 100 | 68246 | 38.26 |
| LogicNet | ex92 | 10 | 3 | 100 | 220936 | 100 | 206941 | 6.33 | 100 | 134192 | 39.26 |
| Average | | | | **100** | 102730 | **100** | 88831 | 13.19 | **100** | **57974** | **45.37** |

Table 2: Experiment results of circuit generation size. Impr. represents the percentage decrease in search space and used NAND gates.

| Benchmark | | CircuitNN | | T-Net | | | | CircuitAR-2B | | | |
|---|---|---|---|---|---|---|---|---|---|---|---|
| | | # NAND↓ | | # NAND↓ | | Impr.(%)↑ | | # NAND↓ | | Impr.(%)↑ | |
| Category | IWLS | Search Space | Used | Search Space | Used | Search Space | Used | Search Space | Used | Search Space | Used |
| Random | ex00 | 700 | 58 | 400 | 68 | 42.86 | -17.24 | 126 | 61 | 82.00 | -5.17 |
| | ex01 | 700 | 66 | 400 | 62 | 42.86 | 6.06 | 138 | 66 | 80.29 | 0.00 |
| Basic Functions | ex11 | 300 | 52 | 180 | 52 | 40.00 | 0.00 | 98 | 45 | 67.33 | 13.46 |
| | ex16 | 700 | 78 | 400 | 59 | 42.86 | 24.36 | 113 | 57 | 83.86 | 26.92 |
| | ex17 | 800 | 109 | 500 | 98 | 37.50 | 10.09 | 196 | 95 | 75.50 | 12.84 |
| Expresso | ex38 | 800 | 98 | 500 | 94 | 37.50 | 4.08 | 178 | 86 | 77.75 | 12.24 |
| | ex46 | 800 | 77 | 500 | 78 | 37.50 | -1.30 | 161 | 79 | 79.88 | -2.60 |
| Arithmetic Functions | ex50 | 300 | 59 | 180 | 56 | 40.00 | 5.09 | 77 | 48 | 74.33 | 18.64 |
| | ex53 | 1000 | 118 | 600 | 116 | 40.00 | 1.70 | 185 | 111 | 81.50 | 5.93 |
| LogicNet | ex92 | 1000 | 99 | 600 | 90 | 40.00 | 9.09 | 168 | 86 | 83.20 | 13.13 |
| Average | | 710 | 81.40 | 426 | 77.30 | 40.11 | 4.19 | **144** | **73.40** | **78.56** | **9.54** |

**Data Scaling.** Figure 6 illustrates consistent performance gains with increased training tokens across all sizes. For the 2B model, BitsD improves by 8.13%, 0.4748→0.4362, when scaling from 1B to 4B tokens. Moreover, larger models exhibit superior data efficiency. Specifically, the 2B model achieves better performance with 4B tokens than the 1B model, emphasizing the interplay between model capacity and training scale.

**Emergent Capability.** Figure 7 highlights CircuitAR's emergent capability in generating complex circuit structures. A clear phase transition is observed at the 2B parameter threshold, where circuit depth increases significantly compared to the 1B model, indicating an emergent capacity for handling structural complexity. Moreover, an inverse correlation between model scale and NAND gate count reveals an efficiency paradigm. Although models with small parameters maintain similar component counts, the 2B model achieves a reduction in NAND gates despite its increased depth, suggesting enhanced topological optimization capabilities at scale. This emergent behavior demonstrates that increasing model parameters can enhance structural efficiency in circuit generation.

## 4.3 SOTA CIRCUIT GENERATION

Given the superior performance of CircuitAR-2B, as demonstrated in Section 4.2, we employ it to generate preliminary circuit structures conditioned on truth tables, which are subsequently refined using DAS. Detailed experimental results are presented in Table 1 and Table 2.

**Efficiency.** As illustrated in Table 1, CircuitAR-2B achieves a 45.37% average improvement in optimization steps compared to CircuitNN, while maintaining 100% accuracy according to the provided truth tables. This performance significantly surpasses T-Net's 13.19% improvement. The

substantial reduction in optimization steps indicates that the preliminary circuit structures generated by CircuitAR-2B effectively prune the search space without compromising the quality of DAS.

**Effectiveness.** Table 2 demonstrates that CircuitAR-2B reduces NAND gate usage by an average of 9.54% compared to CircuitNN, while simultaneously reducing the search space by 78.56%. Notably, for both basic functions (e.g., ex16, with a 26.92% reduction) and complex benchmarks (e.g., ex92, with a 13.13% reduction), our method exhibits superior hardware resource utilization compared to the baseline approaches. This dual improvement in search space compression and circuit compactness underscores the effectiveness of the preliminary circuit structures generated by our CircuitAR-2B under the condition of the truth tables.

Critically, the 100% accuracy across all benchmarks confirms that our method maintains functional correctness while achieving these efficiency gains. This is guaranteed by the DAS process. Specifically, the training does not terminate until the loss converges to a near-zero threshold. At this point, the generated circuit is functionally equivalent to the target truth table, ensuring perfect accuracy. This is not merely an empirical observation but a direct result of the rigorous optimization process in DAS, which enforces logical correctness by design. These results validate our hypothesis that integrating learned structural priors with CircuitAR enables more efficient circuit generation compared to CircuitNN (Hillier et al., 2023b) and template-driven (Wang et al., 2024) DAS approaches.

**Circuit Size.** Compared to prior probabilistic generative models (Li et al., 2024b; Tsaras et al., 2024; Zhou et al., 2024), our method achieves an order-of-magnitude improvement in directly generatable circuit scale, which is a significant advance, especially given the exponential complexity growth typical in the scaling of circuit size. In practical circuit optimization (Hillier et al., 2023a), large circuits are typically partitioned into smaller subcircuits for tractable optimization. Our primary focus is to explore the direct capabilities of generative models in circuit generation. Current evaluation allows us to evaluate the core contributions of our approach without focusing on the additional complexity of decomposition and reintegration.

## 4.4 ABLATION STUDY

We conducted an ablation study to evaluate the effectiveness of the probability matrix $\hat{\mathcal{A}}$ generated by CircuitAR-2B. As summarized in Section 4.4, the experiment results reveal that both variants achieve 100% accuracy across all benchmarks, suggesting that the initialization process does not impair the ability to generate functionally equivalent circuits. The primary distinction lies in the efficiency of the search process, quantified by the number of search steps. Section 4.4 underscores the significance of the initialization process, demonstrating that our CircuitAR models can produce high-quality preliminary circuit structures, which can guide the subsequent DAS process effectively.

Figure 8: Ablation study on the initialization with edge probability generated by CircuitAR-2B.

| Benchmark | | CircuitAR-2B w/o init | | CircuitAR-2B | |
|---|---|---|---|---|---|
| Category | IWLS | Acc.(%)↑ | Steps↓ | Acc.(%)↑ | Steps↓ |
| Random | ex00 | 100 | 72364 | 100 | 52023 |
| | ex01 | 100 | 40528 | 100 | 29636 |
| Basic Functions | ex11 | 100 | 64517 | 100 | 47231 |
| | ex16 | 100 | 76066 | 100 | 45434 |
| | ex17 | 100 | 88609 | 100 | 58548 |
| Expresso | ex38 | 100 | 99594 | 100 | 74847 |
| | ex46 | 100 | 40892 | 100 | 26854 |
| Arithmetic Function | ex50 | 100 | 69958 | 100 | 42729 |
| | ex53 | 100 | 89627 | 100 | 68246 |
| LogicNet | ex92 | 100 | 162651 | 100 | 134192 |
| **Average** | | **100** | 80481 | **100** | **57974** |

## 5 CONCLUSION

Our work introduces a novel approach for circuit generation that combines conditional generative models with DAS. We begin by training CircuitVQ, a circuit tokenizer, and then use it with CircuitAR, a masked autoregressive model. CircuitAR generates preliminary circuit structures from truth tables, which are then refined by DAS to produce functionally equivalent circuits. This approach shows the potential of masked autoregressive models for structured data and offers a new framework for graph generation in other specialized fields.

## DECLARATION OF LLM USAGE

The usage of LLMs is strictly limited to aid and polish the paper writing.

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

# A    RELATED WORKS

## A.1    AUTOREGRESSIVE MODELING

The autoregressive modeling paradigm (OpenAI, 2023; Tian et al., 2024) has been widely adopted for generation tasks in language and vision domains. Built on the transformer architecture, autoregressive models are commonly implemented using causal attention mechanisms in language domains (OpenAI, 2023), which process data sequentially. However, information does not inherently require sequential processing in vision and graph generation tasks. To address this, researchers have employed bidirectional attention mechanisms for autoregressive modeling (Li et al., 2024a; Tian et al., 2024; Chang et al., 2022; Li et al., 2023b). This approach predicts the next token based on previously predicted tokens while allowing unrestricted communication between tokens, thereby relaxing the sequential constraints of traditional autoregressive methods. In this paper, we adopt masked autoregressive modeling for circuit generation, leveraging its ability to provide a global perspective and enhance the modeling of complex dependencies.

## A.2    CIRCUIT GENERATION

In addition to the DAS-based approaches, researchers have also explored next-gate prediction techniques inspired by LLMs for circuit generation. Circuit Transformer (Li et al., 2024b) predicts the next logic gate using a depth-first traversal and equivalence-preserving decoding. SeaDAG (Zhou et al., 2024) employs a semi-autoregressive diffusion model for DAG generation, maintaining graph structure for precise control. ShortCircuit (Tsaras et al., 2024) uses a transformer to generate Boolean expressions from truth tables via next-token prediction. However, these existing methods are fundamentally limited by their global view, which restricts the scalable generation of larger circuits and fails to reduce the search space adequately. Consequently, these models are typically limited to generating circuits with an average size of 20 to 30 nodes (gates). In contrast, our approach utilizes global-view masked autoregressive decoding to generate circuits while simultaneously ensuring logical equivalence and significantly reducing the search space during the DAS process. This method effectively handles circuits up to approximately 100 nodes, representing an order-of-magnitude leap in the complexity that generative models can tackle.

# B    IMPLEMENTATION DETAILS

## B.1    CIRCUITVQ

The training process of CircuitVQ employs a linear learning rate schedule with the AdamW optimizer set at a learning rate of $2 \times 10^{-4}$, a weight decay of 0.1, and a batch size of 2048. The model is fine-tuned for 20 epochs on 8×A100 GPUs with 80G memory each. Moreover, we use the Graphormer (Ying et al., 2021) as our CircuitVQ architecture, as mentioned before. Specifically, CircuitVQ comprises 6 encoder layers and 1 decoder layer. The hidden dimension and FFN intermediate dimension are 1152 and 3072, respectively. Additionally, the multi-head attention mechanism employs 32 attention heads. For the vector quantizer component, the codebook dimensionality is set to 4 to improve the codebook utilization, and the codebook size is configured to 8192.

## B.2    CIRCUITAR

The training process of CircuitAR employs a linear learning rate schedule with the AdamW optimizer set at a learning rate of $2 \times 10^{-4}$, a weight decay of 0.1, and a batch size of 4096. The model is fine-tuned for 20 epochs on 16×A100 GPUs with 80G memory each. Moreover, we use the Trans-

Table 3: Model architecture configurations of CircuitAR.

|  | hidden dim | FFN dim | heads | layers |
|---|---|---|---|---|
| CircuitAR-300M | 1280 | 3072 | 16 | 16 |
| CircuitAR-600M | 1536 | 4096 | 16 | 20 |
| CircuitAR-1B | 1800 | 6000 | 24 | 24 |
| CircuitAR-2B | 2048 | 8448 | 32 | 30 |

former variant of Llama (Dubey et al., 2024) as our CircuitAR architecture as mentioned before. To form different model sizes, we vary the hidden dimension, FFN intermediate dimension, number of heads and number of layers. We present the details of our CircuitAR architecture configurations in Table 3. For the rest of the hyperparameters, we keep them the same as the standard Llama model.

### B.3    TRUTH TABLE REPRESENTATION

A constraint of PI/PO $\leq 15$ is imposed during the training phase to effectively manage the exponential growth in the size of the truth table representations. Given that practical circuit optimization frequently relies on decomposition into smaller sub-circuits, we contend that restricting the generative model's scope to PI/PO $\leq 15$ is a pragmatic design choice. This limitation ensures that the resulting synthesized sub-circuits remain at a size manageable by CircuitAR.

### B.4    DAS IMPLEMENTATION

The search space for the baselines is a hyperparameter-dependent space defined by architectural constraints (e.g., maximum depth and width). Determining these architectural hyperparameters, which are often informed by prior knowledge (rectangular for CircuitNN, trapezoidal for TNet), requires significant computational effort before the primary search can commence. Consequently, the metric reported in Table 2 for these baselines represents the size of the search space that successfully yielded the optimal architecture across five runs. In contrast, our proposed approach offers a significant advantage by minimizing this hyperparameter search cost. We simply provide CircuitAR with a fixed budget (e.g., 1000 gates) to generate the initial predefined circuit structure. For our method, the search space metric is straightforwardly defined as the number of NAND gates in this CircuitAR-predefined initial circuit. This dramatic reduction in the reliance on costly hyperparameter tuning is a key benefit we aim to illustrate.

As for the DAS step, the DAS is terminated upon reaching a predefined hard limit of $1000000$ steps. Alternatively, the DAS is considered successful and stopped early when the loss (defined as the functional error between the generated circuit and the target truth table) decreased to $1 \times 10^{-12}$. This threshold indicates a circuit that is fully functionally equivalent to the target design.

## C    TRAINING DATASETS

This section presents a multi-output subcircuit extraction algorithm designed for generating training datasets. The algorithm processes circuits represented in the And Inverter Graph (AIG) format by iterating over each non-PI node as a pivot node. The extraction process consists of three key stages:

1. **Single-Output Subcircuit Extraction.** The algorithm extracts single-output subcircuits by analyzing the transitive fan-in of the pivot node. The transitive fan-in includes all nodes that influence the output of the pivot node, encompassing both direct predecessors and nodes that propagate signals to it. The extraction process employs a breath-first search (BFS) algorithm, constrained by a maximum input size, to ensure comprehensive coverage of relevant nodes associated with the pivot node.

2. **Multi-Output Subcircuit Expansion.** Single-output subcircuits are expanded into multi-output subcircuits through transitive fan-out exploration. The transitive fan-out comprises all nodes influenced by the pivot node, including immediate successors and downstream nodes reachable through signal propagation. This expansion captures the broader network of nodes that either influence or are influenced by the subcircuits of the pivot node.

3. **Truth Table Generation.** The algorithm computes truth tables for the extracted subcircuits to serve as training labels. Additionally, these truth tables help identify functionally equivalent subcircuits. Recognizing these equivalences is essential, as it can lead to data imbalance in the training set.

To mitigate data imbalance, a constraint is imposed, limiting each truth table to at most $M$ distinct graph topologies. For truth tables with fewer than $M$ representations, logic synthesis techniques (specifically rewriting algorithms) are applied to generate functionally equivalent subcircuits with

Table 4: Experiment results of circuit generation accuracy, DAS steps, and circuit generation size using sub-circuits extracted from the IWLS competition.

| | | T-Net | | | CircuitAR-2B | | |
|---|---|---|---|---|---|---|---|
| Category | # Circuits | Acc.(%)↑ | Steps↓ | # NAND↓ | Acc.(%)↑ | Steps↓ | # NAND↓ |
| IWLS Circuits | 100 | 100 | 76096.62 | 49.08 | 100 | 44922.30 | 48.86 |

Table 5: Comparison between T-Net and our CircuitAR-2B using mapped area and timing metrics.

| Benchmark | | | | T-Net | | CircuitAR-2B | |
|---|---|---|---|---|---|---|---|
| Category | IWLS | # PI | # PO | Area $(\mu m^2)\downarrow$ | Delay $(ps)\downarrow$ | Area $(\mu m^2)\downarrow$ | Delay $(ps)\downarrow$ |
| Random | ex00 | 6 | 1 | 31.92 | 96.94 | 38.30 | 94.62 |
| | ex01 | 6 | 1 | 63.84 | 99.27 | 47.35 | 86.30 |
| Basic Functions | ex11 | 7 | 1 | 56.92 | 91.50 | 26.87 | 98.43 |
| | ex16 | 5 | 5 | 43.36 | 82.42 | 27.66 | 62.47 |
| | ex17 | 6 | 6 | 48.94 | 97.04 | 63.57 | 90.55 |
| Expresso | ex38 | 8 | 7 | 57.72 | 102.47 | 62.24 | 85.22 |
| | ex46 | 5 | 8 | 43.36 | 111.08 | 63.57 | 75.60 |
| Arithmetic Function | ex50 | 8 | 2 | 23.14 | 91.04 | 27.13 | 101.52 |
| | ex53 | 8 | 2 | 132.20 | 168.59 | 90.71 | 148.58 |
| LogicNet | ex92 | 10 | 3 | 93.10 | 106.36 | 53.73 | 156.87 |
| Average | | | | 59.45 | 104.67 | 50.11 | 100.02 |

distinct topologies. This approach ensures topological diversity while maintaining functional equivalence. Finally, the training datasets with around 400000 circuits (average 200 gates per circuit) are generated using circuits from the ISCAS'85 (Bryan, 1985), IWLS'05 (Albrecht, 2005), and EPFL (Amarú et al., 2015). $M$ is set to 5 during the generation process. The sizes of PI and PO are capped at 15 each in the training dataset, ensuring manageable truth table sizes while maintaining complexity.

## D  DATA AUGMENTATION

Following dataset generation, we identified that the data volume is still insufficient. To address this limitation, we implemented data augmentation techniques. Leveraging the topological invariance of graphs, we randomly shuffle the order of graph nodes, as this operation does not alter the underlying structure of the circuit. Furthermore, since inserting idle nodes preserves the circuit structure, we randomly introduce idle nodes into the graphs. The proportion of idle nodes is randomly selected, ranging from 0% to 80%. Moreover, incorporating idle nodes enables CircuitAR to identify which nodes can remain inactive for a fixed number of nodes. This allows CircuitAR to generate circuits logically equivalent to the truth table while utilizing fewer graph nodes. This strategy can improve CircuitAR's efficiency and enhance its generalizability.

## E  SUPPLEMENTARY EXPERIMENTS

### E.1  EXTENSIVE EXPERIMENTS

To significantly expand our evaluation set, we extract 100 diverse sub-circuits from various large-scale designs. These 100 sub-circuits are sampled by selecting the PIs/POs based on the truth tables derived from the IWLS competition (Mishchenko et al., 2022). As demonstrated in Table 4, our proposed CircuitAR substantially reduces the number of steps required in the DAS process and consequently accelerates the overall search time compared to baseline methods. This further validates the efficiency and generalizability of the proposed generative pipeline.

### E.2  MAPPED RESULTS

Considering that mapped area and timing metrics are the ultimate concerns for designers, we provide a comparison between T-Net and our CircuitAR-2B using mapped area and timing metrics. Specif-

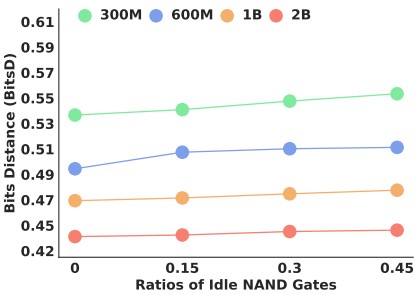

Figure 9: The impact of idle NAND gates on BitsD for CircuitAR with different ratios of isolated NAND gates.

ically, technology mapping is performed by the mapper in the logic synthesis tool ABC (Brayton & Mishchenko, 2010) with the NanGate 45-nm technology library (fre), using a standard mapping and optimization flow that incorporates gate-resizing refinement and generates the final timing and area metrics. As shown in Table 5, CircuitAR-2B demonstrates marginally superior area and timing performance compared with T-Net, achieved by utilizing fewer DAS search steps.

## F  IDLE NAND GATES

As shown in Figure 9, all models exhibit a gradual decline in BitsD as the isolated gates proportion increases from 0% to 45%. Large model with 2B parameters demonstrates significantly greater robustness, maintaining BitsD values within a narrow range across varying isolation ratios. In contrast, the small model with 300M parameters shows a more pronounced degradation, with BitsD increasing from 0.5317 to 0.5484 under the same conditions. This disparity highlights the enhanced ability of larger models to efficiently utilize NAND gates for implementing the same truth table. The consistently low BitsD observed in the 2B model underscores its practical utility in predefining search spaces for DAS, offering a notable advantage over smaller models.

## LIMITATIONS

This study presents a preliminary investigation of scaling laws under current computational constraints. Due to limited computing resources, the research is intentionally bound to models operating within restricted model parameters and training data. Consequently, while the BitsD metric effectively serves as a heuristic for search efficiency (DAS steps), correlating with fewer required steps for functional convergence, we do not observe a direct linear correlation on this limited scale with final circuit metrics. Although our experimental framework demonstrates a tenfold increase in circuit complexity compared to prior works (Li et al., 2024b; Zhou et al., 2024; Tsaras et al., 2024), there is substantial potential for further improvement in circuit scale.

