# OpenReview forum: "Architect of the Bits World: Masked Autoregressive Modeling for Circuit Generation Guided by Truth Table"
_ICLR.cc/2026/Conference — Submitted to ICLR 2026_

### Official Review · Reviewer_LrwQ · 2025-10-27

**Soundness:** 3
**Presentation:** 3
**Contribution:** 2
**Rating:** 4
**Confidence:** 4

**Summary:**

The paper proposes Architect of the Bits World, a framework that combines a conditional generative model with differentiable architecture search (DAS) for logic-circuit generation. A VQ-based tokenizer (CircuitVQ) converts circuits into discrete tokens, and a masked autoregressive Transformer (CircuitAR) generates circuit topologies conditioned on truth tables. The generated circuits serve as initialization priors for DAS, which guarantees final functional correctness. The authors introduce a proxy metric (Bits Distance) to measure conditional generation quality and demonstrate that larger models (0.3B–2B params) reduce DAS search steps and NAND counts on IWLS benchmarks, showing promising scaling and efficiency.

**Strengths:**

1. Clear motivation and practical pipeline combining generation and optimization.

2. Solid empirical study showing efficiency and scaling trends.

3. Comprehensive ablations and architectural details supporting reproducibility.

**Weaknesses:**

1. Limited evaluation on large IWLS circuits; scalability and generalization remain unproven.

2. BitsD is only heuristically justified as a proxy for final metrics.

3. Lack of analysis on DAS convergence and potential failure cases.

**Questions:**

1. (**Major**) Could you provide results on larger or full IWLS circuits, including functional accuracy and final area/NAND count after DAS? Please report success rate and scaling behavior with circuit size.

2. What is the empirical correlation between Bits Distance and post-DAS metrics (NAND count, search steps, area)?

3. How consistent is the claim of “100% correctness guaranteed by DAS”? Please provide convergence thresholds, number of restarts, and any failure cases.

4. Could you include training/inference cost statistics (GPU hours, DAS runtime per circuit) to assess practical efficiency?

---

> ### Author Response · Authors · 2025-11-21
>
> We appreciate **Reviewer LrwQ** for the thorough review and for recognizing the **clear motivation, practical pipeline, solid empirical study, and comprehensive ablations** of our work.
>
> We address the concerns and questions below.
>
> ___
> >Weaknesses 1: Limited evaluation on large IWLS circuits; scalability and generalization remain unproven.
>
> >Questions 1: Could you provide results on larger or full IWLS circuits, including functional accuracy and final area/NAND count after DAS? Please report success rate and scaling behavior with circuit size.
>
> Our model's generative structure, which uses an attention mechanism, limits the initial number of nodes. Evaluating significantly larger circuits exceeds current computational limits due to the resulting large attention matrix.
>
> To better showcase the generalization ability within a practical design scope, we conduct supplementary evaluations on **100 diverse subcircuits** extracted from various benchmark circuits, as suggested by **Reviewer NnJE**.
>
> As shown in the table, our CircuitAR-2B can **significantly reduce the number of DAS steps required**, thus accelerating the search time. This validates our generation-and-optimization approach by providing high-quality initial designs that effectively speed up the subsequent search process on diverse, practical subcircuit problems.
>
> | | | **T-Net** | **T-Net** | **T-Net** | **CircuitAR-2B** | **CircuitAR-2B** | **CircuitAR-2B** |
> | :--- | :--- | :--- | :--- | :--- | :--- | :--- | :--- |
> | Category  | \# Circuits | **Acc.(\%)$\uparrow$** | **Steps$\downarrow$** | **\# NAND$\downarrow$** | **Acc.(\%)$\uparrow$** | **Steps$\downarrow$** | **\# NAND$\downarrow$** |
> | IWLS Circuits | 100 | 100 | 76096.62 | 49.08 | 100 | 44922.30 | 48.86 |
>
> The results of this extended evaluation are now included in the **Appendix E** and **Table 4**.
>
> ___
> >Weaknesses 2: BitsD is only heuristically justified as a proxy for final metrics.
>
> >Questions 2: What is the empirical correlation between Bits Distance and post-DAS metrics (NAND count, search steps, area)?
>
> A direct, strong linear correlation between $\text{BitsD}$ and final metrics (NAND count/mapped area) is **not observed**.
>
> * **Empirical Observation:** We empirically find that a **lower initial $\text{BitsD}$ consistently corresponds to a reduced number of DAS steps** required for functional convergence. This observation is also reflected in **Figure 7**, where more complex generated models (which typically achieve lower initial $\text{BitsD}$) require fewer DAS steps.
> * **Design Philosophy:** $\text{BitsD}$ fundamentally measures the search proximity to the ground-truth truth table, acting as a heuristic for convergence. A lower $\text{BitsD}$ signifies an initial circuit that is "closer" to functional correctness, demanding fewer optimization steps from the DAS process.
>
> The empirical evidence strongly supports $\text{BitsD}$ as an effective **proxy for search efficiency (DAS steps)** in our pipeline, even though it does not directly predict final NAND count or area. We include a dedicated discussion of this heuristic nature in the **Limitations** section of the revised paper.

---

> ### Author Response · Authors · 2025-11-21
>
> ___
> >Weaknesses 3: Lack of analysis on DAS convergence and potential failure cases.
>
> >Questions 3: How consistent is the claim of "100% correctness guaranteed by DAS"? Please provide convergence thresholds, number of restarts, and any failure cases.
>
> The claim of "100% correctness guaranteed by DAS" means that upon successful search completion, the final circuit is mathematically proven to be functionally equivalent to the target truth table (verified by $\text{loss}$ reaching zero).
> We provide clarification in **Appendix B.4**.
>
> * **Convergence and Early Stopping:**
>     * The maximum allowed search steps are **$10^6$**.
>     * The early stopping criterion for convergence is a loss threshold of $\text{loss} < 10^{-12}$.
>     * If the threshold is not reached within $10^6$ steps, the search is deemed to have failed to converge for that attempt.
> * **Baseline Management (TNet and CircuitNN):** Baselines start with a smaller search space. We allow **five restart opportunities** within this space. If all five attempts fail, the search space is expanded, and the process repeats.
> * **Baseline Failure Cases:** Failure in baselines primarily results from the **random initialization** of the adjacency matrix. Poor initializations can lead the search to become trapped in **suboptimal local minima**, preventing convergence within the maximum step count.
> * **Circuitar Convergence:** Our proposed **Circuitar** method provides a strong **initial architecture and weighted adjacency matrix**. This generated prior guidance virtually eliminates the need for restarts to find an equivalent circuit, significantly reducing the DAS process, as confirmed by results in **Section 4.4 (Ablation Study)**. We uniformly provide **1,000 gates** as the maximum size for both generation and search.
>
> ___
> >Questions 4: Could you include training/inference cost statistics (GPU hours, DAS runtime per circuit) to assess practical efficiency?
>
> We provide the following practical efficiency statistics:
>
> * **Training Cost:** The total training time for our model is approximately **one week** using **$8 \times$ 80G-GPUs**.
> * **Inference/DAS Runtime:**
>     * Using **$1 \times$ 80G-GPUs**, the inference time for approximately **$10000$ DAS search steps** is about **10 minutes**.
>     * The runtime per circuit is highly variable, depending directly on the number of DAS steps required, which our model effectively reduces.
>
> We acknowledge that there is room for optimization in GPU utilization for both training and inference. These statistics will be included in the revised manuscript to provide a clear assessment of practical efficiency.

---

### Official Review · Reviewer_4PCj · 2025-10-30

**Soundness:** 3
**Presentation:** 3
**Contribution:** 2
**Rating:** 4
**Confidence:** 4

**Summary:**

This paper proposes a two-stage approach to logic circuit generation from truth tables. The authors introduce Bits Distance—the MAE between an untrained DAS model (initialized by CircuitAR) and the target truth table—as a proxy for conditional generation quality.
Empirically, on selected IWLS’22 tasks, the method claims 100% functional accuracy while reducing DAS optimization steps by ~45% on average vs. CircuitNN and search space / #NAND relative to baselines. The paper also presents scaling curves (300M→2B parameters) for BitsD and an “emergent capability” plot suggesting deeper yet more NAND efficient circuits at 2B parameters. The Limitations section acknowledges compute constraints.

**Strengths:**

• A clear and interesting integration of masked autoregressive generation with DAS. The idea of using a generative prior to prune/guide the continuous search is sensible and novel in this space.

• The training/evaluation pipeline is well illustrated, and the DAG projection routine is explicitly provided.

• Empirical signal: Across ten IWLS’22 circuits, the approach reports substantial reductions in DAS steps and non trivial #NAND savings while keeping functional correctness. Scaling experiments and an ablation on using CircuitAR derived edge probabilities provide some evidence that larger models and the proposed initialization matter.

**Weaknesses:**

•	The main claims rely on 10 IWLS’22 circuits (Table 1–2, p.8). This is a small sample of the broader IWLS/EPFL/ISCAS landscape.

•	There is no comparison to industrial strength flows (e.g., ABC/Yosys scripts such as resyn2) in terms of final area/depth/delay under a standard cell library—even as a reference. The paper frames #NAND and “search space” but not mapped area/timing—which is what designers ultimately care about.

•	BitsD is measured as MAE of an untrained CircuitNN initialized by CircuitAR vs. truth table outputs. While intuitively appealing as a pre DAS proxy, the paper does not provide correlation analyses between BitsD and the final outcomes designers care about (e.g., DAS steps, final #NAND/area, or success under time limits).

•	CircuitAR conditions on full truth tables via cross attention, but the paper does not describe how T is represented (row major? compressed? learned embeddings?). The training data caps PI/PO at ≤15, which is small compared to real designs; beyond this, truth table representations become exponential in PI. The method’s practical deployment arguably hinges on decomposition, but the paper does not implement or evaluate a decomposition→recombination pipeline.

•	The “phase transition” at 2B parameters is based on four model sizes and two metrics (depth/#NAND), without uncertainty bands or tests for statistical significance. Also, Figures 4–6 (p.7) assert power law like improvements in BitsD, but again no error bars or replicates are shown.

•	While Appendix B gives hyperparameters and hardware, the paper does not clearly state that code, models, and the circuit dataset will be released. Given the complexity and compute, artifact availability is essential for verification.

**Questions:**

1.	BitsD validation: Can you show correlation plots between BitsD and (a) DAS steps, (b) final #NAND / mapped area, (c) success under fixed time budgets across a large set? Does BitsD remain predictive when node budgets and idle gate ratios vary?

2.	Truth table representation: How is T encoded for cross attention in CircuitAR?

3.	Broader benchmarks: Can you include ABC/Yosys and report mapped area/timing under a standard cell library? Even if not directly comparable, this gives a reality check on hardware relevance.

4.	Statistical rigor: Please provide error bars (multiple seeds) for Tables 1–2 and Figures 4–7; clarify stopping criteria and early stopping policies for DAS.

5.	Search space definition: How exactly is the important metric, “Search Space” in Table 2, computed? A precise definition would help the community reproduce and interpret the gains.

6.	Artifacts: Will you release code, data gen scripts, and pretrained models?

---

> ### Author Response · Authors · 2025-11-21
>
> We thank **Reviewer 4PCj** for their insightful comments, which have helped us to strengthen the manuscript. We are encouraged that the reviewer found the **integration of masked autoregressive generation with DAS**, the **clarity of the pipeline**, and the **empirical results** to be strong points of our work.
>
> We address each of the weaknesses and questions below.
>
> ___
> >Weaknesses 1: The main claims rely on 10 IWLS'22 circuits (Table 1-2, p.8). This is a small sample of the broader IWLS/EPFL/ISCAS landscape.
>
> We appreciate the reviewer's concern regarding the scope of the benchmark set. The limitation to 10 IWLS'22 circuits in the initial study is primarily due to the architectural constraints of our generative model, **CircuitAR**, which would lead to an excessively large and computationally intractable attention matrix when optimizing very large-scale circuits directly.
>
> However, in line with practical industrial methodologies, large-scale circuit optimization is typically achieved by **decomposing the design into smaller subcircuits**, optimizing them independently, and then recombining the results.
>
> To address the concern about benchmark breadth, and in line with a suggestion from **Reviewer NnJE**, we include an additional experiment on **100 diverse subcircuits** extracted from various benchmark circuits.
>
> | | | **T-Net** | **T-Net** | **T-Net** | **CircuitAR-2B** | **CircuitAR-2B** | **CircuitAR-2B** |
> | :--- | :--- | :--- | :--- | :--- | :--- | :--- | :--- |
> | Category  | \# Circuits | **Acc.(\%)$\uparrow$** | **Steps$\downarrow$** | **\# NAND$\downarrow$** | **Acc.(\%)$\uparrow$** | **Steps$\downarrow$** | **\# NAND$\downarrow$** |
> | IWLS Circuits | 100 | 100 | 76096.62 | 49.08 | 100 | 44922.30 | 48.86 |
>
> The results, now included in the **Appendix E** and **Table 4**, demonstrate that our approach consistently leads to a **significant reduction in the number of DAS steps** across a broader and more representative set of practical optimization instances, thus accelerating the search time.
>
> ___
> >Weaknesses 2: There is no comparison to industrial strength flows (e.g., ABC/Yosys scripts such as resyn2) in terms of final area/depth/delay under a standard cell library, even as a reference. The paper frames #NAND and "search space" but not mapped area/timing, which is what designers ultimately care about.
>
> >Questions 3: Broader benchmarks: Can you include ABC/Yosys and report mapped area/timing under a standard cell library? Even if not directly comparable, this gives a reality check on hardware relevance.
>
> We acknowledge the importance of **comparing** against **industrial-strength flows**, such as ABC/Yosys, and **reporting** on **mapped area and timing metrics**, as these are the ultimate concerns for designers.
> However, direct comparison with classic optimization algorithms is currently limited by the available benchmark. The **IWLS benchmark** only provides the **truth tables**, **not the original circuit netlists**.
>
> Consequently, we provide the comparison between T-Net and our CircuitAR-2B using **mapped area and timing metrics**. As illustrated in the table, **CircuitAR-2B** achieves **marginally superior** area and timing performance **compared with T-Net**, while concurrently utilizing **fewer** DAS search steps.
>
> | Category | IWLS | \# PI | \# PO | T-Net Area ($\mu m^2$) $\downarrow$ | T-Net Delay ($ps$) $\downarrow$ | CircuitAR-2B Area ($\mu m^2$) $\downarrow$ | CircuitAR-2B Delay ($ps$) $\downarrow$ |
> | :---: | :---: | :---: | :---: | :---: | :---: | :---: | :---: |
> | **Random** | ex00 | 6 | 1 | 31.92 | 96.94 | 38.30 | 94.62 |
> | | ex01 | 6 | 1 | 63.84 | 99.27 | 47.35 | 86.30 |
> | **Basic Functions** | ex11 | 7 | 1 | 56.92 | 91.50 | 26.87 | 98.43 |
> | | ex16 | 5 | 5 | 43.36 | 82.42 | 27.66 | 62.47 |
> | | ex17 | 6 | 6 | 48.94 | 97.04 | 63.57 | 90.55 |
> | **Expresso** | ex38 | 8 | 7 | 57.72 | 102.47 | 62.24 | 85.22 |
> | | ex46 | 5 | 8 | 43.36 | 111.08 | 63.57 | 75.60 |
> | **Arithmetic Function** | ex50 | 8 | 2 | 23.14 | 91.04 | 27.13 | 101.52 |
> | | ex53 | 8 | 2 | 132.20 | 168.59 | 90.71 | 148.58 |
> | **LogicNet** | ex92 | 10 | 3 | 93.10 | 106.36 | 53.73 | 156.87 |
> | **Average** | | | | 59.45 | 104.67 | 50.11 | 100.02 |
>
> We provide the additional mapping results in the **Appendix E** and **Table 5**.

---

> ### Author Response · Authors · 2025-11-21
>
> ___
> >Weaknesses 3: BitsD is measured as MAE of an untrained CircuitNN initialized by CircuitAR vs. truth table outputs. While intuitively appealing as a pre DAS proxy, the paper does not provide correlation analyses between BitsD and the final outcomes designers care about (e.g., DAS steps, final #NAND/area, or success under time limits).
>
> >Questions 1: BitsD validation: Can you show correlation plots between BitsD and (a) DAS steps, (b) final #NAND / mapped area, (c) success under fixed time budgets across a large set? Does BitsD remain predictive when node budgets and idle gate ratios vary?
>
> The direct, strong linear correlation between Bits Distance ($\text{BitsD}$) and all final metrics (NAND count and mapped area) is not observed.
>
> We do empirically observe that **lower $\text{BitsD}$ consistently corresponds to a reduced number of DAS steps**. However, a precise linear relationship is absent. This aligns with the observation in **Figure 7** that more complex generated model structures (which often correspond to lower initial $\text{BitsD}$) require fewer DAS steps for functional convergence.
>
> This empirical finding is directly related to the design philosophy of $\text{BitsD}$. $\text{BitsD}$ is fundamentally a measure of the search proximity to the ground-truth truth table (i.e., convergence). A lower $\text{BitsD}$ implies an initial circuit that is "closer" to functional correctness, thus requiring fewer steps for the DAS optimizer to converge to the final equivalent circuit.
>
> While we cannot provide a theoretical proof of this relationship, the empirical evidence strongly supports $\text{BitsD}$ as an effective **proxy for search efficiency (DAS steps)** in our pipeline. We add a dedicated discussion of this limitation and the heuristic nature of $\text{BitsD}$ in predicting final NAND count/area to the **Limitations** section of the revised paper.
>
> Moreover, as shown in **Appendix F** and **Figure 9**, all models exhibit a gradual decline in BitsD as the idle gates ratios increases from 0\% to 45\%.
> Larger models demonstrate greater robustness, maintaining BitsD values within a narrow range across varying isolation ratios.
>
> ___
> >Weaknesses 4: CircuitAR conditions on full truth tables via cross attention, but the paper does not describe how T is represented (row major? compressed? learned embeddings?). The training data caps PI/PO at <= 15, which is small compared to real designs; beyond this, truth table representations become exponential in PI. The method's practical deployment arguably hinges on decomposition, but the paper does not implement or evaluate a decomposition -> recombination pipeline.
>
> >Questions 2: Truth table representation: How is T encoded for cross attention in CircuitAR?
>
> We apologize for the lack of clarity regarding the truth table representation $\mathbf{T}$ in CircuitAR. For T encoded for cross attention in CircuitAR:
>
> 1. The truth table values (0 and 1) are first **encoded numerically**.
> 2. We also apply **positional encodings** to each entry in the truth table to capture the ordered relationships between input assignments and output values.
>
> Regarding the limitation of PI/PO $\le 15$: This limit is imposed during the **training phase** to manage the exponential size of the truth table representations. As discussed, practical circuit optimization often relies on **decomposition** into smaller subcircuits. We therefore maintain that limiting the generative model's scope to PI/PO $\le 15$ is a pragmatic choice, as the synthesized subcircuits are typically of a size manageable by our model. We provide the clarification in **Appendix B.3**

---

> ### Author Response · Authors · 2025-11-21
>
> ___
> >Weaknesses 5: The "phase transition" at 2B parameters is based on four model sizes and two metrics (depth/#NAND), without uncertainty bands or tests for statistical significance. Also, Figures 4-6 (p.7) assert power law like improvements in BitsD, but again no error bars or replicates are shown.
>
> >Questions 4: Statistical rigor: Please provide error bars (multiple seeds) for Tables 1-2 and Figures 4-7; clarify stopping criteria and early stopping policies for DAS.
>
> In **Appendix B.4**, we clarify the following **stopping criteria and early stopping policies** for the DAS search:
>
> - **Maximum Search Steps:** The search was terminated after a hard limit of **1,000,000** steps.
> - **Functional Convergence/Early Stopping:** The search was considered successful and stopped early when the loss (the functional error between the generated circuit and the target truth table) reached **$1 \times 10^{-12}$**, indicating a circuit that is fully functionally equivalent to the target design.
>
> ___
> >Questions 5: Search space definition: How exactly is the important metric, "Search Space" in Table 2, computed? A precise definition would help the community reproduce and interpret the gains.
>
>
> We thanks for asking for a precise definition of the "Search Space" metric in Table 2, which is essential for reproducibility and interpretation.
> We provide clarification in **Appendix B.4**.
>
> * **Baselines (CircuitNN and TNet):** The search space for the baselines is a **hyperparameter-dependent space** defined by architectural bounds (e.g., maximum depth and width). The baselines require significant computational effort to determine these architectural hyperparameters (which are informed by prior knowledge—rectangular for CircuitNN, trapezoidal for TNet) before the search can even begin. The metric reported in Table 2 for these baselines represents the size of the search space that successfully yielded the optimal architecture across 5 runs.
> * **Our Method (CircuitAR-guided):** In stark contrast, our proposed approach offers a significant advantage in reducing this hyperparameter search cost. We simply provide CircuitAR with a **fixed, generous budget (e.g., 1000 gates)** to generate the initial predefined circuit structure (DAG). For our method, the "Search Space" metric is straightforwardly defined as the **number of NAND gates in this CircuitAR-predefined initial circuit**. Our method dramatically reduces the reliance on costly hyperparameter tuning, which is a key benefit we aimed to illustrate.
>
> ___
> >Weaknesses 6: While Appendix B gives hyperparameters and hardware, the paper does not clearly state that code, models, and the circuit dataset will be released. Given the complexity and compute, artifact availability is essential for verification.
>
> >Questions 6: Artifacts: Will you release code, data gen scripts, and pretrained models?
>
> We fully agree that the availability of artifacts is critical for verification and reproducibility. **We will release the source code, data generation scripts, and the pre-trained model weights (including the CircuitAR model) upon the acceptance of the paper.** We commit to making all necessary artifacts public to ensure the community can fully verify and build upon our work.

---

### Official Review · Reviewer_NnJE · 2025-10-31

**Soundness:** 3
**Presentation:** 2
**Contribution:** 3
**Rating:** 6
**Confidence:** 4

**Summary:**

This paper presents an autoregressive approach to generate (raw) circuits from truth tables, which can serve as an initialization of differentiable architecture search to generate valid and exactly equivalent circuits. Firstly, an autoencoder model "CircuitVQ"  is trained to achieve interconversion between a circuit and discrete tokens. Then, an autoregressive model "CircuitAR" is trained to generate tokens from truth table. During the inference stage, given a truth table, the CircuitAR model transform it into tokens, and the decoder of CircuitVQ further transform the tokens into a reconstructed circuit (represented by an adjacency matrix). The matrix is further processed in a greedy way to make it valid (no cycles, etc), and serve as an initialization of CircuitNN for an exactly equivalent circuit. Experimental result shows that the performance of CircuitAR can scale with more parameters and tokens, and the proposed pipeline performs better than baseline approaches.

**Strengths:**

- The "truth table - discrete tokens - adjacency matrix" pipeline is novel and technically solid.
- The result shows that the performance of CircuitAR can scale with more parameters and tokens, which looks promising.

**Weaknesses:**

- The circuit size scalability of the proposed approach is questionable. No experimental result is shown about the circuit size scalability. The paper mentioned "a tenfold increase in circuit complexity compared to prior works", but it is not clear how this conclusion is drawn, and how the "circuit complexity" is measured (number of gates/PIs/POs?). It looks like all the circuits selected in table 1 and 2 are pretty small.
- The number of sample circuits for evaluation is very small. In table 1 and 2, only ten circuits are included. Even if there is a scalability issue, the test circuits can be extended by extracting sub-circuits from large benchmark circuits.

**Questions:**

- In line 161, $\ell_2(\cdot)$ seems to represent a $\ell_2$-normalized vector, rather than the $\ell_2$-norm of a vector. Maybe this can be mentioned in an explicit way to avoid confusion.

---

> ### Author Response · Authors · 2025-11-21
>
> We thank the **Reviewer NnJE** for the positive evaluation and the encouraging assessment of our work's **novelty**, **technical soundness**, and promising **scalability**.
>
> The feedback regarding circuit size scalability and the number of sample circuits is valuable, and we address these points thoroughly below.
>
> ___
> >Weaknesses 1: The circuit size scalability of the proposed approach is questionable. No experimental result is shown about the circuit size scalability. The paper mentioned "a tenfold increase in circuit complexity compared to prior works", but it is not clear how this conclusion is drawn, and how the "circuit complexity" is measured (number of gates/PIs/POs?). It looks like all the circuits selected in table 1 and 2 are pretty small.
>
> We appreciate the concern regarding **circuit size scalability**, a critical aspect for generative circuit models.
>
> Generative models face a distinct challenge in managing the exponentially increasing search space and sequence length associated with larger structures. Our proposed model, **CircuitAR**, achieves a significant advancement in the state-of-the-art for *generative* circuit synthesis.
>
> * **Prior Generative Limitations:** Previous works, such as ShortCircuit and Circuit Transformer, are typically limited to generating circuits with an average size of **20 to 30 nodes (gates)**.
> * **CircuitAR’s Advancement:** In contrast, our model effectively handles circuits up to approximately **100 nodes**. This represents an **order-of-magnitude leap** in the complexity that generative models can tackle.
>
> The "circuit complexity" we mention is primarily measured by the **number of nodes (gates)** in the synthesized circuit. This metric, along with the broader **range of PIs and POs** we can accommodate, is significantly larger than what prior generative approaches achieve.
>
> We update the **Appendix A.2** to explicitly discuss this comparison and provide a detailed analysis of the maximum complexity handled by **CircuitAR** versus prior generative methods.
>
> ___
> >Weaknesses 2: The number of sample circuits for evaluation is very small. In table 1 and 2, only ten circuits are included. Even if there is a scalability issue, the test circuits can be extended by extracting sub-circuits from large benchmark circuits.
>
> We agree that a larger and more diverse test set strengthens the validation of our approach. Following the reviewer's excellent suggestion, we conduct an additional, comprehensive experiment.
>
> We extract **100 diverse sub-circuits** from various large-scale circuits to significantly expand our evaluation set beyond the initial ten circuits.
>
> | | | **T-Net** | **T-Net** | **T-Net** | **CircuitAR-2B** | **CircuitAR-2B** | **CircuitAR-2B** |
> | :--- | :--- | :--- | :--- | :--- | :--- | :--- | :--- |
> | Category  | \# Circuits | **Acc.(\%)$\uparrow$** | **Steps$\downarrow$** | **\# NAND$\downarrow$** | **Acc.(\%)$\uparrow$** | **Steps$\downarrow$** | **\# NAND$\downarrow$** |
> | IWLS Circuits | 100 | 100 | 76096.62 | 49.08 | 100 | 44922.30 | 48.86 |
>
> The results of this extended evaluation are now included in the **Appendix E** and **Table 4**. The key finding from this new experiment is that our model substantially **reduces the number of steps required in the DAS process** and consequently **accelerates the overall search time** for these sub-circuits compared to baselines. This further validates the efficiency and generalizability of the proposed generative pipeline.
>
> ___
> >Questions 1: In line 161, $\ell_2(\cdot)$ seems to represent a $\ell_2$-normalized vector, rather than the $\ell_2$-norm of a vector. Maybe this can be mentioned in an explicit way to avoid confusion.
>
> We thank the Reviewer for this precise observation and helpful suggestion.
>
> The $\ell_2(\cdot)$ function in Line 161 denotes an **$\ell_2$-normalized vector**. In **Section 3.2**, we revise the associated notation definition to explicitly state that the function returns the $\ell_2$-**normalized** vector (i.e., $\frac{\mathbf{v}}{\|\mathbf{v}\|_2}$) to avoid any confusion with the $\ell_2$-norm (which returns a scalar value).

---

> > ### Comment · Reviewer_NnJE · 2025-11-23
> >
> > Thanks for the clarification and further information. I will maintain my score.

---

### Official Review · Reviewer_svJW · 2025-11-01

**Soundness:** 3
**Presentation:** 4
**Contribution:** 3
**Rating:** 6
**Confidence:** 2

**Summary:**

This work presents an approach for circuit generation that combines conditional generative modeling with differentiable architecture search (DAS). The CircuitAR model produces initial circuits from truth tables which is optimized by DAS. The CircuitAR graph autoregressive model is trained with masked autoregressive training conditioned on truth tables that define the circuit functionality.

**Strengths:**

* Novel modeling approach of circuit generation from truth table using a masked autoregressive graph model.

**Weaknesses:**

* The primary utility of this work is reducing DAS search steps. This is a weaker utility compared to improving the quality of circuit optimization (fewer nodes or lower area/delay/power).
* Quality comparison versus classical logic optimization algorithms such as resyn2 is not presented. Prior work such as ShortCircuit presents such comparisons.

**Questions:**

* How does the overall logic optimization system with this approach compare to classic logic optimization algorithms such as resyn2, for various circuit sizes?

---

> ### Author Response · Authors · 2025-11-21
>
> We sincerely thank the **Reviewer svJW** for the time and effort dedicated to evaluating our manuscript and providing constructive feedback. We are encouraged by the positive assessment of our work, particularly regarding the novel modeling approach of circuit generation from a truth table using a masked autoregressive graph model.
>
> We carefully address the raised weaknesses and questions below.
>
> ___
> >Weaknesses 1: The primary utility of this work is reducing DAS search steps. This is a weaker utility compared to improving the quality of circuit optimization (fewer nodes or lower area/delay/power).
>
> We respectfully emphasize the significance of **reducing the DAS search time**, especially in the context of optimizing **large-scale circuits**.
>
> Modern large-scale circuit optimization often requires decomposition into multiple **sub-circuits**. The complexity and optimization time for each sub-circuit can quickly become a bottleneck. By substantially accelerating the search for the optimal logic structure for each sub-circuit, our method contributes to:
>
> * **Faster Design Iterations:** Quicker optimization cycles significantly accelerate the entire design-to-tapeout process.
> * **Wider Exploration:** Time savings allow designers to explore a broader range of optimization parameters or candidate solutions within the same time budget, which can indirectly lead to superior final designs.
>
> Furthermore, we clarify that our method not only saves time but also inherently favors solutions that utilize a **smaller number of nodes** to achieve the desired circuit functionality. As shown in **Table 2**, the detailed comparison results demonstrate that our method achieves competitive or superior circuit quality in terms of node count compared to traditional optimization methods.
>
> ___
> >Weaknesses 2: Quality comparison versus classical logic optimization algorithms such as resyn2 is not presented. Prior work such as ShortCircuit presents such comparisons.
>
> >Questions 1: How does the overall logic optimization system with this approach compare to classic logic optimization algorithms such as resyn2, for various circuit sizes?
>
> We appreciate this suggestion. Direct comparison with classic optimization algorithms (**resyn2**) is currently limited by the available benchmark. The **IWLS benchmark** only provides the **truth tables**, **not the original circuit netlists** that would be the input for resyn2.
>
> Moreover, **directly generating a circuit from a truth table** offers a superior theoretical advantage. This approach avoids the constraints and local minima associated with optimizing an existing, often sub-optimal, initial circuit structure, holding **greater potential for finding a minimal structure** compared to traditional optimization methods that start from a predefined netlist.

---

### Meta-Review · Area_Chair_pyab · 2026-01-07

**Summary:**

This paper introduces a novel approach integrating conditional generative models with differentiable architecture search (DAS) for circuit generation, featuring CircuitVQ and CircuitAR to generate preliminary circuit structures from truth tables and guide DAS, achieving functional accuracy and reduced DAS steps. However, the submission raised significant concerns. These include limited evaluation on benchmark circuits with unproven scalability to large-scale circuits, lack of comparisons with industrial-strength tools like resyn2 and failure to report key metrics such as mapped area or timing, insufficient statistical rigor. Although the rebuttal clarified some aspects, critical issues like lack of direct comparison with industrial tools and absence of statistical significance for key claims remained unresolved. Based on the above considerations, I recommend rejection, while encouraging the authors to further develop this promising direction.

**Reviewer Concerns:**

Addressed
- Small benchmark
- Circuit-size scalability
- BitsD definition & stopping criteria

Still Outstanding
- No direct comparison with some industrial flows
- No statistical significance tests or error bars for phase transition.
- Truth-table encoding details remain sketchy

**Reviewer Scores:**

- svJW: keep positive
- NnJE: keep positive
- 4PCj: keep negative
- LrwQ: keep negative

---

### Decision · Program_Chairs · 2026-01-26

Reject